# Transcriptional Effects of Psychoactive Drugs on Genes Involved in Neurogenesis

**DOI:** 10.3390/ijms21218333

**Published:** 2020-11-06

**Authors:** Chiara C. Bortolasci, Briana Spolding, Srisaiyini Kidnapillai, Timothy Connor, Trang T.T. Truong, Zoe S.J. Liu, Bruna Panizzutti, Mark F. Richardson, Laura Gray, Michael Berk, Olivia M. Dean, Ken Walder

**Affiliations:** 1The Institute for Mental and Physical Health and Clinical Translation, Barwon Health, Deakin University, Geelong 3220, Australia; briana.spolding@deakin.edu.au (B.S.); timothy.connor@deakin.edu.au (T.C.); truongtra@deakin.edu.au (T.T.T.T.); zoe.liu@deakin.edu.au (Z.S.J.L.); b.panizzuttiparry@deakin.edu.au (B.P.); l.gray@deakin.edu.au (L.G.); michael.berk@deakin.edu.au (M.B.); o.dean@deakin.edu.au (O.M.D.); ken.walder@deakin.edu.au (K.W.); 2School of Medicine, Centre for Molecular and Medical Research, Deakin University, Geelong 3220, Australia; srisaiyini.kidnapillai@med.lu.se; 3School of Life and Environmental Sciences, Genomics Centre, Deakin University, Geelong 3220, Australia; m.richardson@deakin.edu.au; 4Department of Psychiatry, Royal Melbourne Hospital, University of Melbourne, Parkville 3052, Australia; 5Centre of Youth Mental Health, University of Melbourne, Parkville 3052, Australia; 6Orygen Youth Health Research Centre, Parkville 3052, Australia; 7Florey Institute for Neuroscience and Mental Health, University of Melbourne, Parkville 3052, Australia

**Keywords:** neurogenesis, psychotropic drugs, neurons, bipolar disorder, schizophrenia, psychiatry, mental health, neuroscience

## Abstract

Although neurogenesis is affected in several psychiatric diseases, the effects and mechanisms of action of psychoactive drugs on neurogenesis remain unknown and/or controversial. This study aims to evaluate the effects of psychoactive drugs on the expression of genes involved in neurogenesis. Neuronal-like cells (NT2-N) were treated with amisulpride (10 µM), aripiprazole (0.1 µM), clozapine (10 µM), lamotrigine (50 µM), lithium (2.5 mM), quetiapine (50 µM), risperidone (0.1 µM), or valproate (0.5 mM) for 24 h. Genome wide mRNA expression was quantified and analysed using gene set enrichment analysis, with the neurogenesis gene set retrieved from the Gene Ontology database and the Mammalian Adult Neurogenesis Gene Ontology (MANGO) database. Transcription factors that are more likely to regulate these genes were investigated to better understand the biological processes driving neurogenesis. Targeted metabolomics were performed using gas chromatography-mass spectrometry. Six of the eight drugs decreased the expression of genes involved in neurogenesis in both databases. This suggests that acute treatment with these psychoactive drugs negatively regulates the expression of genes involved in neurogenesis in vitro. *SOX2* and three of its target genes (*CCND1*, *BMP4*, and *DKK1*) were also decreased after treatment with quetiapine. This can, at least in part, explain the mechanisms by which these drugs decrease neurogenesis at a transcriptional level in vitro. These results were supported by the finding of increased metabolite markers of mature neurons following treatment with most of the drugs tested, suggesting increased proportions of mature relative to immature neurons consistent with reduced neurogenesis.

## 1. Introduction

Neurogenesis is the development of new neurons from neural stem/precursor cells (NSCs). In the adult human brain, the NSCs reside primarily in two regions: The subgranular zone in the dentate gyrus of the hippocampus, and the subventricular zone of the lateral ventricles [1].

The NSCs are a diverse population of cells with the capacity to self-renew and differentiate into neurons in response to stimuli in order to maintain central nervous system homeostasis. Local stimuli from the niche where these cells are found, as well as extrinsic factors such as neurotransmitters, growth factors, cytokines, and adhesion molecules, can positively or negatively affect the NSCs’ state and differentiation potential, thereby modulating neurogenesis in the adult brain [1]. For example, microglial activation and secretion of pro-inflammatory cytokines are processes involved with healthy aging in the brain, and this pro-inflammatory environment is deleterious to NSCs, thereby decreasing neurogenesis [2]. Age-related changes in the cell cycle due to accumulation of damaged proteins can cause a reduction in the proliferation rate of NSCs [3] and cause the cells to become quiescent, but they can be reactivated upon stimulation, such as by physical exercise or epileptic seizures, through Notch signalling activation [4].

This form of circuit plasticity may be altered in different diseases and targeted with pharmacological therapies [5]. Cognitive deficits, mood dysregulation, and declines in hippocampal volume have been correlated with mental disorders that present decreased neurogenesis, such as major depression, post-traumatic stress disorder, schizophrenia, and Alzheimer’s disease [6]. Neurogenesis in the hippocampus is necessary for the therapeutic effects of antidepressants [7], but in the subventricular zone of the lateral ventricles, for example, chronic treatment with fluoxetine decreased neurogenesis [8]. However, if chronic stress was first induced (known to be both a contributor to the development of mood disorders [9] and decreased neurogenesis), the treatment with fluoxetine and imipramine can revert the decline in neurogenesis by increasing the NSCs pool and survival [10].

While the effects of antidepressants acting through serotonergic receptors is widely investigated and thus better understood, the effects and mechanisms of action of other psychoactive drugs in neurogenesis remain unknown and/or controversial, with studies in mice showing positive effects on neurogenesis. Chronic treatment with atypical antipsychotics (olanzapine, quetiapine, clozapine, risperidone, and aripiprazole) increased neurogenesis in the hippocampus of adult mice [11]. In contrast, haloperidol treatment resulted in decreased neurogenesis [11]. On the other hand, in humans, the use of atypical antipsychotics (risperidone, olanzapine, paliperidone, amisulpride, and aripiprazole) was associated with a reduction in grey matter volume in first-episode schizophrenia patients [12], suggesting a reduction in neurogenesis.

The changes that occur in the NSCs in response to environmental cues such as stress, psychiatric disorders, and aging, together with the evidence of NSCs responsiveness to drugs used in treating psychiatric disorders, supports the idea of neurogenesis being a potential therapeutic target for these diseases. Therefore, this study aims to evaluate the effects of common psychoactive drugs, used in the treatment of affective disorders (bipolar disorder and schizophrenia), on the expression of genes involved in neurogenesis in a human neuronal model in cell culture.

## 2. Results

### 2.1. Gene Ontology (GO) Database

The effects of the eight individual psychoactive drugs on the expression level of genes in the GO database classified as involved in neurogenesis were investigated. The effects were quantified and expressed as the enrichment score (ES) and the normalised ES (NES) of all genes following gene set enrichment analysis (GSEA). The results are summarised in Table 1, and the GSEA plot for each drug is shown in Appendix A.

As shown in Table 1, 7 out of the 8 drugs of interest appeared to cause a general decrease in the expression of genes involved in neurogenesis (as indicated by negative ES and NES). Among the 7 drugs associated with reduced gene expression, 6 showed significant *p* values (*p* < 0.05). Risperidone was the only drug that led to an increase in the expression of genes involved in neurogenesis.

Having established that the drug treatments generally decreased the expression of genes involved in neurogenesis based on the GO database, we further identified the number of genes that were differentially expressed after the drug treatments.

As illustrated in Figure 1, risperidone and lamotrigine were shown to regulate the least number of differentially expressed genes individually (*n* = 0 and *n* = 1, respectively) whilst quetiapine appeared to regulate the most (*n* = 586). Interestingly, a number of differentially expressed genes were commonly regulated by more than 2 and up to 5 of the drugs.

### 2.2. Mammalian Adult Neurogenesis Gene Ontology (MANGO) Database

The GO database contains 1818 genes that are annotated as being involved in neurogenesis. However, it is likely that a number of these genes are only peripherally involved, many of which are not supported by empirical evidence, and may have more important roles in other biological processes. Therefore, we sought to identify an alternative ontological database that was more specific for genes that play key roles in neurogenesis. The MANGO database classifies approximately 250 genes as involved in neurogenesis and provides experimental evidence to accurately define the role of each gene. Effects of the drugs on MANGO neurogenesis genes are shown in Table 2. Because the data were not normally distributed, the change in gene expression level for each gene was quantified and expressed as a median log fold change (logFC) with 95% confidence intervals for the median value.

The results showed that 6 of the 8 drugs significantly down-regulated the expression of genes involved in neurogenesis as classified in MANGO (amisulpride, aripiprazole, clozapine, lamotrigine, lithium, and quetiapine). Among the various drugs, quetiapine (median logFC = −0.053), clozapine (−0.028), and lithium (−0.023) showed the strongest effects. Overall, these findings are consistent with the previous results generated using the GO database.

To compare the effects on the expression levels of genes involved in the various processes of neurogenesis across 6 of the 8 drugs (lamotrigine and risperidone were excluded from the analysis due to empty counts), a χ^2^ test was performed. The result showed no statistically significant differences between the drugs of interest (*p* = 0.87). As depicted in Figure 2, the lack of statistical significance is likely due to the fact that the drugs showed similar patterns of proportions for the various processes that comprise neurogenesis, as defined by MANGO. The complete list of these unique genes and the processes they are categorised in are shown in Appendix A.

To visualise the number of unique genes transcriptionally regulated by the different drug treatments, both individually and collectively, Figure 3 was plotted using the MANGO database neurogenesis genes with the FDR cut-off set at <0.05. In order to focus on the commonalities between various drugs, we applied a more stringent standard for differentially expressed genes to be considered as overlapping, i.e., genes must be differentially regulated in the same direction (negative or positive trend) across multiple drugs.

Similar to the findings reported from using the GO database, risperidone and lamotrigine individually regulated the least number of genes (*n* = 0), whilst quetiapine appeared to regulate the most (*n* = 143). Some genes were shown to be transcriptionally regulated in the same direction by up to 4 drugs. We looked closely at the neurogenesis genes that were differentially regulated by 3 or more of the drugs, as these may represent key targets for future drug development. Table 3 below summarises the data for the 15 genes that are regulated in the same direction by 3 or more of the drugs. For several of these genes that were differentially expressed following drug treatment (FDR < 0.05), we conducted a quantitative reverse transcription polymerase chain reaction (RT-qPCR) to validate these effects. In most cases, we observed a dose-dependent effect of the drugs on the expression of these genes, which confirmed the results obtained from the next generation sequencing dataset (Figure 4). Appendix A contains the entire list of transcription factors that could affect the expression of genes in the MANGO database as assessed using TRRUST.

### 2.3. SOX2

Among the genes regulated by multiple drugs was *SOX2* (Sry-related HMG box 2), a transcription factor previously implicated in neural development and neurogenesis. As shown in Figure 5, SOX2 gene expression was decreased following treatment with quetiapine (*p* = 1.75 × 10^−8^, FDR *q* = 2.76 × 10^−7^), clozapine (*p* = 0.00021, FDR *q* = 0.010), and valproate (*p* = 0.0036, FDR *q* = 0.022), and tended to be decreased by amisulpride (*p* = 0.0027, FDR *q* = 0.088) and aripiprazole (*p* = 0.0072, FDR *q* = 0.11). To validate these effects, RT-qPCR was performed, and the results were compared to the next generation sequencing data. As shown in Figure 6, we observed a dose-dependent effect of clozapine and quetiapine on *SOX2* gene expression. This confirms the next generation sequencing results for these two drugs.

To further investigate the effects of reducing *SOX2* gene expression on neurogenesis genes, we first verified the impact of *SOX2* on the MANGO neurogenesis genes using the TRRUST database. To achieve this, the list of MANGO neurogenesis genes was submitted to and analysed using the TRRUST database, which generated a list of transcription factors that could affect the expression of the genes submitted for query. Fisher’s exact test was then performed based on the contingency table of gene counts. The results obtained for *SOX2* revealed a *p* value of 0.00059, and an FDR *q*-value of 0.00096, indicating that *SOX2* was likely a major regulator of these neurogenesis genes.

As SOX2 is a transcription factor, we then investigated whether the expression of down-stream target genes of SOX2 were affected by the psychotropic agents. We focused on data obtained from the quetiapine, clozapine, and valproate treatment arms as they showed the strongest evidence of negatively regulating SOX2 gene expression compared to the other drugs.

Among the list of genes submitted for query in TRRUST, 3 target genes were identified as most strongly associated with SOX2: Dickkopf WNT signalling pathway inhibitor 1 (*DKK1*), Bone morphogenetic protein 4 (*BMP4*), and Cyclin D1 (*CCND1*). Table 4 below shows the change in next generation sequencing expression levels in these 3 genes following drug treatment. The findings suggest that quetiapine had an overall negative effect on the expression of these genes (logFC range = −0.74 to −0.41) and by implication, neurogenesis. RT-qPCR was used to confirm the effects of the drugs on the expression of these genes (Figure 7). In most cases, the differential expression observed in the next generation sequencing data set were confirmed and showed a dose-dependent effect of the drugs on these genes.

### 2.4. Cell Culture Metabolite Profiling

The six drugs with effects on genes involved in neurogenesis were tested for their effects on the levels of metabolites suggested to either have a role in neurogenesis, or as markers of neuronal maturity (Table 5). Generally, the drugs led to an increase in the levels of most of the metabolites evaluated in NT2-N cells, particularly *N*-acetyl-l-aspartic acid (NAA; increased by amisulpride, aripiprazole, clozapine, and lithium), l-glutamic acid (increased by amisulpride, aripiprazole, and lithium), Gamma-Aminobutyric acid (GABA; increased by amisulpride, aripiprazole, clozapine, and quetiapine) and l-glutamine (increased by amisulpride, aripiprazole, and lithium). The exceptions were amisulpride and l-lactic acid (logFC = −0.35, *p* = 0.042), quetiapine and glutathione (logFC = −2.94, *p* = 0.002), and lithium and GABA (logFC = −0.23, *p* = 0.024).

## 3. Discussion

The generation of functional neurons from progenitor cells, neurogenesis, is a continuing process in the hippocampus throughout life, from embryonic development until adulthood [13]. Altered neurogenesis is implicated in diverse neuropsychiatric disorders including depression, schizophrenia, and bipolar disorder [14]. This study showed the effects of psychoactive drugs on genes involved in neurogenesis. Clozapine, amisulpride, aripiprazole, lithium, quetiapine, and lamotrigine all significantly decreased the expression of genes involved in neurogenesis, while risperidone significantly increased the expression of genes involved in neurogenesis. Apart from risperidone, the findings were confirmed when we investigated the effects of these drugs on the expression of neurogenesis genes using a more curated database (MANGO).

The literature around the effects of atypical antipsychotics in neurogenesis is controversial. In rodent models, chronic administration of aripiprazole, quetiapine, and clozapine significantly increased neurogenesis, cell proliferation, and survival [11,15,16,17]. Kim et al. (2015) suggested a reversal of depressive behaviours through preventing degeneration of dopaminergic neuronal cells and enhancing neurogenesis after treatment with aripiprazole [18]. Aripiprazole treatment increased neurite branches in primary cortical neurons derived from mice with dopamine D2 receptor hyperactivity and disrupter in schizophrenia 1 (DISC1) [19]. Clozapine treatment in male Wistar rats subjected to chronic mild stress upregulated adult neurogenesis and neuronal survival, reversing the behavioural effects of chronic stress [20]. Quetiapine augmentation treatment increased hippocampal cell proliferation and neuronal differentiation and improved depressive-like behaviours in treated rats [21].

In humans, some studies of antipsychotic treatment were associated with grey matter loss as shown in imaging studies in patients with schizophrenia, but studies have yielded inconsistent results as explored in a meta-analysis by Fusar-Poli [22]. Wang et al. (2018) showed a significant decrease in grey matter volume in the left parahippocampal gyrus/hippocampus, right temporal pole mid/superior temporal gyrus, right parahippocampal/hippocampus, and right insula after four weeks of treatment with antipsychotics [23]. Guo et al. (2019) also showed a decrease in grey matter volume in the bilateral frontal, temporal, and left parietal brain regions associated with antipsychotic treatment [12]. In both studies, the majority of the treatment group were using several antipsychotics at the same time, and therefore conclusions on the specific effects of individuals drugs could not be made.

Antipsychotics are also prescribed for patients with bipolar disorder, together with mood stabilisers and antidepressants. These drugs have also been associated with structural brain differences in these patients, but data is still inconclusive and difficult to parse from illness effects. In bipolar disorder, total grey matter volume was reduced in individuals treated with atypical antipsychotics [24]. The ENIGMA study reported significantly reduced cortical surface areas of large prefrontal areas in individuals with bipolar disorder taking atypical antipsychotics [25]. However, lithium, considered the first line treatment for bipolar disorder, has been demonstrated to increase neurogenesis in the dentate gyrus of the rodent hippocampus [26,27]. Another study also demonstrated that lithium could promote neuronal differentiation of hippocampal neural progenitor cells both in vitro and in vivo [28]. Yucel et al. (2008) also demonstrated a bilateral increase in hippocampal volume in bipolar disorder patients treated with lithium, but not lamotrigine [29]. Similarly, in other human imaging studies, grey matter volume increased after four weeks of lithium treatment [30] and appeared to be independent of long-term treatment response [31]. Lithium increases brain-derived neurotrophic factor (BDNF) expression and genes associated with neuroprotection, such as *Bcl2* and *Bcl-XL*, and also decreases the expression of pro-apoptotic genes *Bax*, *Bad*, and caspases in rat hippocampal neurons [32]. This effect on apoptosis could, at least in part, be the mechanism involved in the increase of grey matter volume in vivo, rather than effects on neurogenesis.

Overall, drugs analysed in our study appear to increase neurogenesis in rodent models, which is in contrast to the findings from human imaging studies, where the drugs appear to decrease grey matter volume, suggestive of reduced neurogenesis, although it is unclear in some designs whether this relates to illness effects or other lifestyle or medical confounders. Our data is at transcriptional levels. The increases in grey matter demonstrated in imaging studies probably occurred because of neurotrophic effects and could explain the differences in results. To the best of our knowledge, there is no study with a similar approach to ours.

We further investigated possible mechanisms by which these drugs decrease genes involved in neurogenesis in vitro, identifying a candidate target molecule in *SOX2*. SOX2 is a transcription factor and a marker of the nervous system from the beginning of development in many species, and acts to co-ordinate widespread transcriptional regulation of genes involved in neurogenesis. *SOX2* mutations in humans cause defects in the brain, particularly in the hippocampus, involving cognition, movement control, and vision [33]. As identified by TRRUST analysis, three major gene targets of SOX2 in the neurogenesis pathway are *CCND1*, *BMP4*, and *DKK1*. CCND1 promotes neurogenesis in vivo, a role that is not linked to its cell cycle function [34]. Both in vivo and in vitro studies present evidence of the role of BMP4 in the modulation of adult neurogenesis in the hippocampus [35]. DKK1 is a suppressor of neural stem cell proliferation, and studies demonstrate increased self-renewal of neural progenitors and increased generation of immature neurons after the deletion of *DKK1* in adult brains [36]. The downregulation of *BMP4*, *CCDN1*, and *DKK1* by quetiapine could represent a molecular mechanism for the reduction in expression of neurogenesis genes observed in this study after treatment with quetiapine.

The drugs appeared to cause a general increase in a number of metabolites in NT2-N cells that have previously been suggested to be related to the process of neurogenesis. While it may appear that this suggests a positive effect on neurogenesis, the increased levels of these particular sets of metabolites are related to the presence of more mature neurons, with active neuritogenesis, and not the generation of new immature neurons.

For example, decreased levels of NAA is used as a marker of neuronal integrity and is reportedly decreased after whole-brain radiation in rodents, associated with changes in neurotransmission and loss of neuronal viability [37]. The pathological pruning of dendrites has also been associated with a reduction of NAA, and this could contribute to the finding of reduced brain volume associated with reduced levels of NAA in schizophrenia, bipolar disorder, post-traumatic stress disorder, and obsessive-compulsive disorders [38]. The effects seen in this study corroborate previous studies showing that antipsychotics and mood stabilizers increase the levels of NAA, suggesting a therapeutic response acting on increasing neuronal viability [39].

GABA is another example, with four of the drugs increasing the levels of this metabolite in NT2-N cells. GABA agonists reportedly regulate synaptic integration by increasing the numbers and extension of neurites and promoting the survival of existing neurons [40,41].

In summary, the present study demonstrated that psychoactive drugs decrease the expression of genes involved in neurogenesis in neuronal cell culture after 24 h treatment, partially through inhibition of *SOX2* and its targets genes *BMP4*, *CCDN1*, and *DKK1*. It also showed the alterations in the levels of metabolites involved in neuronal health, particularly NAA and GABA.

This study has some limitations that need to be acknowledged. Only one dose of each drug was tested. In addition, we explored acute effects of these drugs, and there is a possibility of a different effect during chronic administration. Our study did not evaluate the effects of these drugs in models of any particular disease state, therefore the nature of the pathophysiologic process of specific diseases might interfere in the effect of these drugs in patients.

## 4. Materials and Methods

### 4.1. Cell Culture

NTera2/cloneD1 (NT2) is a pluripotent cell line used as a model of human neurons due to its ability to differentiate into post-mitotic neurons (NT2-N) following retinoic acid (RA) treatment [42,43]. NT2 cells (CVCL_0034, ATCC, Manassas, VA, USA) were cultured and differentiated into NT2-N cells as previously described [44]. In summary, cells were cultured and expanded in Dulbecco’s modified Eagle’s Medium (DMEM; Life Technologies, Melbourne, Australia) supplemented with 10% foetal bovine serum (FBS; Thermo Fisher Scientific, Melbourne, Australia) and 1% antibiotic/antimycotic solution (Life Technologies). NT2 cells were treated with 10^−5^ M RA (Sigma-Aldrich, Sydney, Australia) for 28 days, with media refreshed every 2–3 days. For experiments, cells (2 × 10^5^ cells/well) were seeded onto 24-well plates coated with 10 μg/mL poly-d-lysine (Sigma-Aldrich) and 10 μg/mL laminin (Sigma-Aldrich). To enrich for a culture of differentiated neuronal cells, mitotic inhibitors (1 µM cytosine and 10 µM uridine; Sigma-Aldrich) were added to the media every 2–3 days for a total of 7 days. The expression of the neuronal markers *NeuroD*, *Tau*, and *GluR* were evaluated using polymerase chain reaction and agarose gel electrophoresis (data not shown). This was used to validate the generation of differentiated cells with a neuron-like phenotype.

### 4.2. Drug Treatments

NT2-N cells were treated with amisulpride (10 µM), aripiprazole (0.1 µM), clozapine (10 µM), lamotrigine (50 µM), lithium (2.5 mM), quetiapine (50 µM), risperidone (0.1 µM), or valproate (0.5 mM) for 24 h (*n* = 4–6 per group). All drugs were purchased from Sigma-Aldrich (Sydney, Australia). Vehicle control cells were treated with an equal volume of Milli-Q water for lithium or valproate controls, 0.1% dimethyl sulfoxide (DMSO) for amisulpride, aripiprazole, clozapine, and risperidone, and 0.2% DMSO for lamotrigine or quetiapine controls. The drugs that were chosen are commonly used in psychiatry and are considered to be mechanistically different. Doses were selected based on previous dose response studies performed in our lab.

### 4.3. Genome-Wide Gene Expression Quantification

Following the 24-h drug treatment, cells were harvested using Trizol and total RNA was extracted using RNeasy^®^ mini kits (Qiagen, Melbourne, Australia). The quality of the extracted RNA was evaluated using an Agilent 2100 Bioanalyzer (Agilent Technologies, Melbourne, Australia). The RNA quantity was determined using a NanoDrop 1000 (Thermo Fisher Scientific, Waltham, MA, USA).

RNAseq libraries were prepared for all samples from 1 µg total RNA using a TruSeq RNA Sample Preparation Kit (Illumina, Victoria, Australia) as per the manufacturer’s instructions. Samples were run on an Illumina HiSeq platform (HiSeq 2500 rapid 50bpSE; 1 flow cell, 2 lanes) to quantify genome wide mRNA expression.

### 4.4. Genome-Wide Gene Expression Analysis

The raw data were obtained in fastq format and processed using the Deakin Genomics Centre RNA-Seq alignment and expression quantification pipeline (https://github.com/m-richardson/RNASeq_pipe). In summary, this involves: Raw read quality filtering and adapter trimming (ILLUMINACLIP:2:30:10:4, SLIDINGWINDOW:5:20, AVGQUAL:20 MINLEN:36) with Trimmomatic v35 [45], and alignment to the reference genome using STAR v2.5 in 2-pass mode (Human genome version GRCh38) [46]. The expression was quantified at the gene level, and individual sample counts were collated into a m × n matrix for differential abundance testing. Normalisation (TMM) and removal of low expressed gene were performed using edgeR [47] in R [48] following the edgeR manual (<1 cpm in *n* samples, where *n* is the number of samples in the smallest group for comparison).

Differential gene expression analysis was assessed using edgeR in R, and statistical significance was corrected for multiple testing using FDR by applying the Benjamini–Hochberg method on the *p*-values. Genes with FDR *q*-values of <0.05 were considered to be differentially expressed.

### 4.5. GO Database

GSEA was deployed using the R package fgsea from Bioconductor [49,50], with gene lists ranked based on log fold change (logFC) and the neurogenesis gene set retrieved from the GO database by the AMIGO web application [51,52]. The following filters were set for neurogenesis GO annotations: “GO:0022008” as accession identification, “*Homo sapiens*” as organism, and “experimental evidence” as evidence. The resulting tables had enrichment scores and *p*-values calculated from 1000 permutations. The UpSet plot was generated using the UpSetR package in R to summarize the overlap of differentially expressed GO neurogenesis genes (FDR cutoff < 0.05) across the 8 drugs [53,54].

### 4.6. MANGO Database

For the purpose of focusing on genes with well-established evidence of involvement in the neurogenesis process, the MANGO database was utilised to identify genes regulated by the drugs of interest. MANGO version 3.2 contains 397 genes curated and classified according to their effects and associated subprocesses in adult neurogenesis [55]. Differential expression of the MANGO neurogenesis genes by each drug were then tested for overall statistical significance. Distribution of logFC data was checked for normality using Kolmogorov–Smirnov tests. Drug treatment groups were compared against their respective controls using independent samples *t*-tests for normally distributed data and Mann–Whitney U tests for data not normally distributed. The 95% confidence interval for the median was obtained by bootstrapping 1000 samples using R [48]. χ^2^ test of homogeneity was used to test if the proportions for subprocesses of neurogenesis (e.g., proliferation, axonogenesis) were homogenous across different drugs. Aiming to unravel the biological processes driving neurogenesis, we further investigated transcription factors that are more likely to regulate MANGO neurogenesis genes using the prediction tool of key transcription factors from TRRUST—a manually curated database for transcriptional regulatory networks [56].

### 4.7. Validation of Genome-Wide Gene Expression Using RT-qPCR

Following the 24-h treatment, cells were harvested, and RNA was extracted using RNeasy^®^ mini kits (Qiagen) and reverse-transcribed to produce cDNA using Maxima H Minus first strand cDNA synthesis kit (Thermo Fisher Scientific) following the manufacturer’s instructions. RT-qPCR was used to measure the expression of specific genes as listed on Table 6. The experiments were carried out in a QuantStudio 3 Real-time PCR system (Thermo Fisher Scientific) using the following protocol: 95 °C for 7 min, 4 cycles of 95 °C for 30 s, and 60 °C for 1 min and then data acquisition, 60 °C for 30 s, 55–95 °C, data acquisition and 20 °C for 10 s. Resultant melt curves were used as an indicator of amplification specificity. The Quant-iT™ OliGreen^®^ ssDNA Assay Kit (Life Technologies) was used to quantify the cDNA concentration in each sample as per the manufacturer’s instructions. Gene expression data was quantified using the ΔΔC_t_ method normalised to the derived cDNA concentration of each sample. Kolmogorov–Smirnov test was used to check data sets for normality of distribution. Levene’s Test was used to determine whether or not equal variances could be assumed between groups. Drug treatment groups were compared against their respective controls using independent sample *t*-tests for normally distributed data and Mann–Whitney U tests for data not normally distributed. Statistical analysis was performed using Statistical Package for the Social Sciences version 22 (SPSS) software. Differences were considered statistically significant when *p* ≤ 0.05.

### 4.8. Cell Culture Metabolite Extraction and Profiling

Cells were treated as described on Section 4.2. After 24 h of treatment, cells were washed twice with 1× phosphate buffered saline (PBS) at 37 °C then snap-frozen by covering the plate in liquid nitrogen. Metabolites were extracted on ice by addition of 250 µL/well of methanol:chloroform (9:1 *v/v*), containing the internal standards, U-13C-sorbitol (4 µM) and 13C5, 15N-valine (4 µM). Cells were scraped and incubated on ice for 10 min. Samples were then centrifuged (5 min, 14,000 rpm, 4 °C) to pellet precipitated proteins, and the supernatants were transferred to fresh 1.5 mL tubes. Briefly, 200 µL of the aqueous extract was dried for analysis in vacuo. An aliquot of each extracted sample was also taken and pooled to create a pooled biological quality control (PBQC) sample. This pooled sample was split into equivalent loadings as for the biological samples and also dried for analysis. These PBQC samples were run at regular intervals throughout the sample sequence. Gas chromatography–mass spectrometry (GC-MS) analysis was performed following methoximation and trimethylsilylation using a Shimadzu GC/MS-TQ8050NX system and analysed in multiple reaction monitoring (MRM) mode using the Shimadzu Smart Metabolites Database containing 521 MRM metabolite targets. Subsequent data analysis was performed in a targeted manner using Shimadzu LabSolutions Insight software (version 3.6). Briefly, target ion areas for polar metabolites contained within the Shimadzu Smart Metabolites database were integrated and output as a data matrix for downstream data analysis. Each detected metabolite was visually inspected and manually integrated if required. This resulted in a highly curated matrix representing the detected metabolites in each sample.

## Figures and Tables

**Figure 1 ijms-21-08333-f001:**
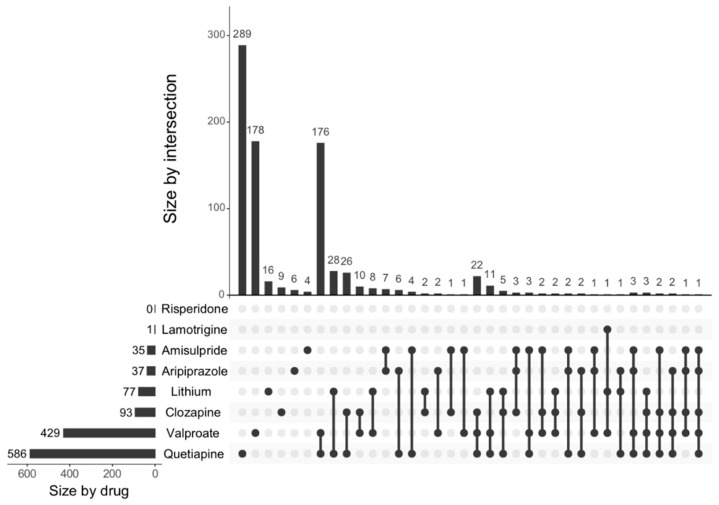
The number of GO neurogenesis related genes regulated by the individual drugs. The intersections of differentially expressed genes from the GO database associated with the administration of 8 drugs are shown. The results were obtained based on the comparison against vehicle-treated cells and the false discovery rate (FDR) cut-off was set at <0.05. The total number of differentially expressed genes regulated by each drug is shown on the leftmost horizontal axis. The top panel shows the vertical axis, which represents the unique sets of differentially expressed genes affected by 1 drug and overlapping sets between multiple drugs. The connections between these sets are indicated by dots and connected lines, as shown in the bottom panel.

**Figure 2 ijms-21-08333-f002:**
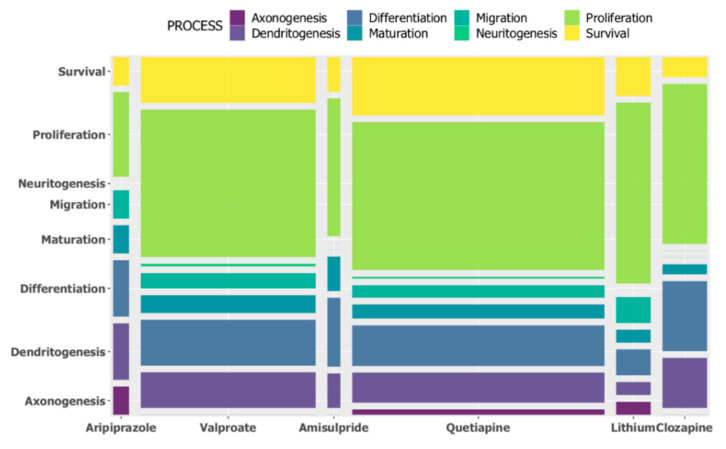
The distribution of proportions for multiple neurogenesis processes for the six drugs shown as a mosaic plot of frequencies. The effects of the six individual drugs on the expression of the MANGO neurogenesis genes involved in the various processes were visualised. Lamotrigine and risperidone were excluded due to empty counts. The width of each bar represents the relative number of genes regulated by each drug on the *x*-axis, and the height is proportional to the number of genes categorised under each process on the *y*-axis.

**Figure 3 ijms-21-08333-f003:**
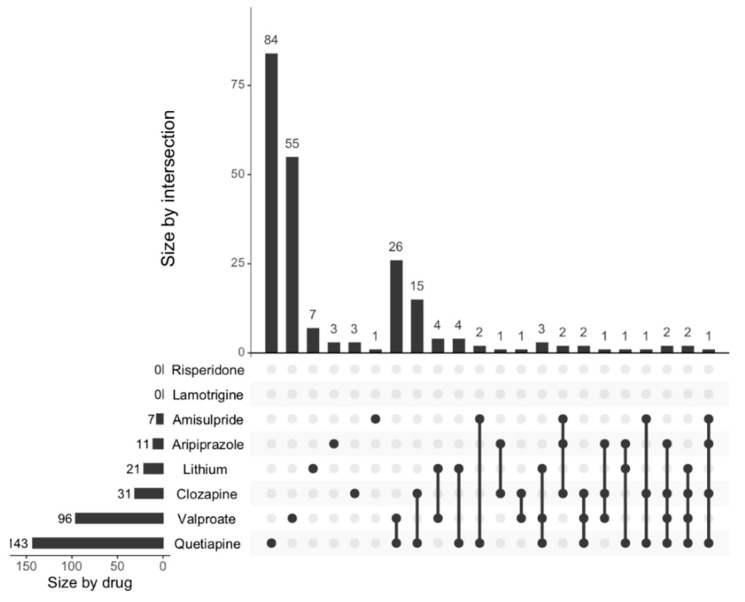
The number of MANGO neurogenesis related genes regulated by the individual drugs. The intersections of differentially expressed genes from the MANGO database associated with the administration of the eight drugs are shown. The results were obtained based on the comparison against vehicle-treated cells, and the FDR cut-off was set at <0.05. The total number of differentially expressed genes regulated by each drug is shown on the leftmost horizontal axis. The top panel shows the vertical axis, which represents the unique sets of differentially expressed genes affected by 1 drug and overlapping sets between multiple drugs. The connections between these sets are indicated by dots and connected lines as shown in the bottom panel.

**Figure 4 ijms-21-08333-f004:**
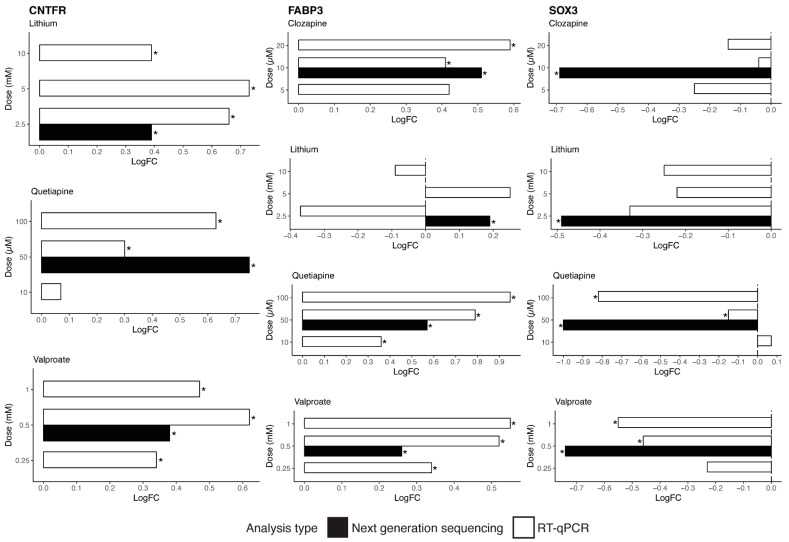
The effects of the drugs at various doses on the regulation of *CNTFR*, *FABP3*, and *SOX3* as revealed by next generation sequencing and RT-qPCR data.* *p* < 0.05.

**Figure 5 ijms-21-08333-f005:**
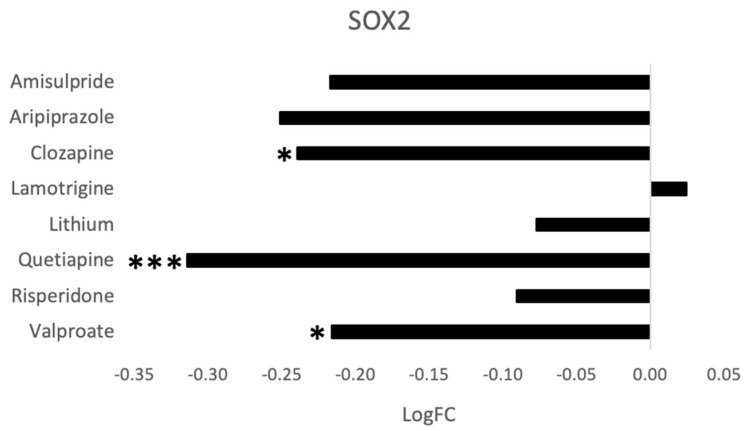
The effects of the eight drugs on *SOX2* gene expression. * *p* < 0.05, *** FDR *q* < 0.05.

**Figure 6 ijms-21-08333-f006:**
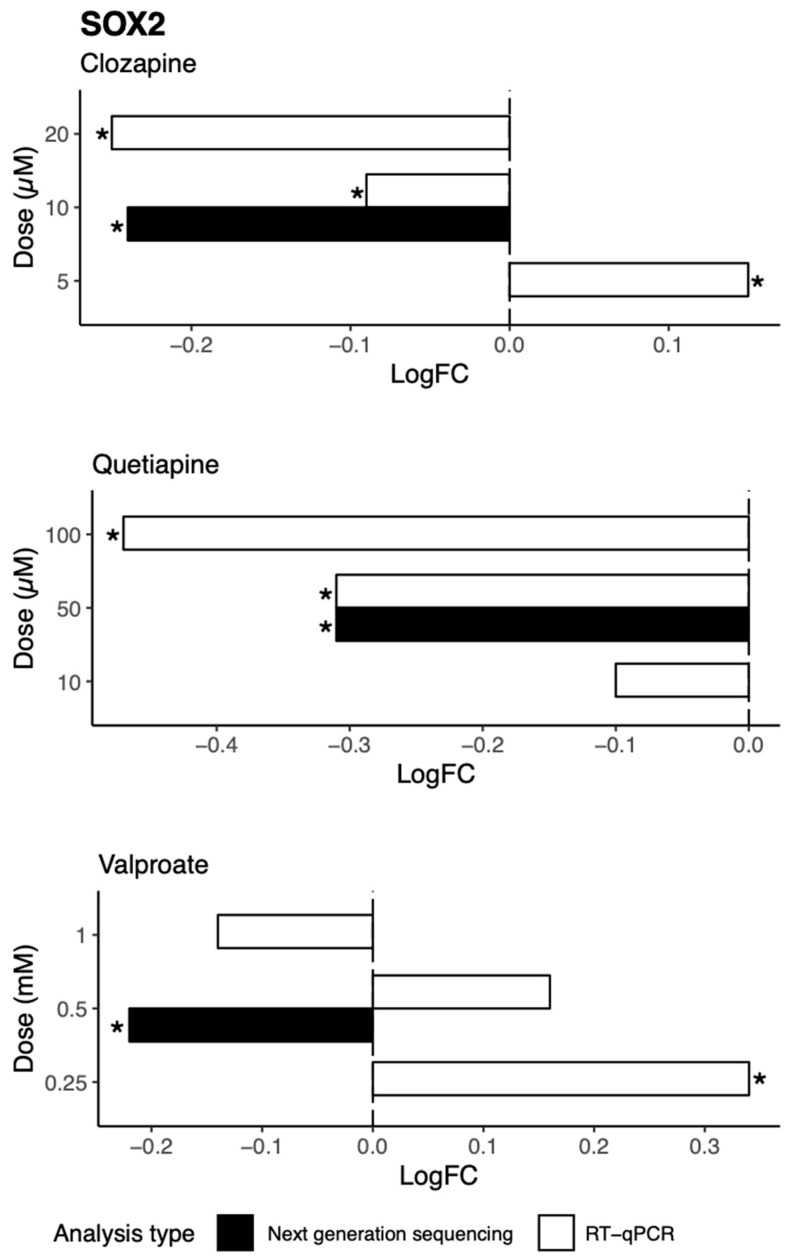
The effects of clozapine, quetiapine, and valproate at various doses on the regulation of *SOX2* as revealed by next generation sequencing and RT-qPCR data. * *p* < 0.05.

**Figure 7 ijms-21-08333-f007:**
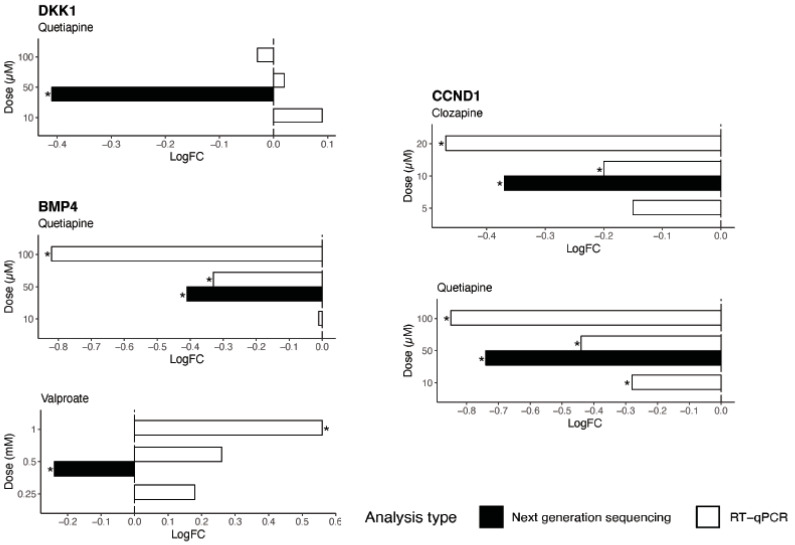
The effects of the drugs at various doses on the regulation of 3 target genes of *SOX2* as revealed by next generation sequencing and RT-qPCR data. * *p* < 0.05.

**Table 1 ijms-21-08333-t001:** The effects of the drugs on Gene Ontology (GO) neurogenesis gene expression.

Drug	*p* Value	ES	NES
Amisulpride	0.016	−0.087	−1.93
Aripiprazole	0.0021	−0.11	−2.44
Clozapine	0.0021	−0.13	−2.86
Lamotrigine	0.0020	−0.10	−2.19
Lithium	0.0021	−0.16	−3.65
Quetiapine	0.0082	−0.089	−1.96
Risperidone	0.014	0.084	1.81
Valproate	0.086	−0.064	−1.42

Abbreviations: ES = Enrichment score; NES = Normalised enrichment score.

**Table 2 ijms-21-08333-t002:** The effects of the drugs on Mammalian Adult Neurogenesis Gene Ontology (MANGO) neurogenesis gene expression.

Drug	MANGO Neurogenesis Genes		
	Median	95% CI	*p* Value
Amisulpride	−0.014	−0.047/0.001	0.010
Aripiprazole	−0.014	−0.034/−0.001	0.008
Clozapine	−0.028	−0.049/−0.012	<0.001
Lamotrigine	−0.011	−0.022/0.000	0.016
Lithium	−0.023	−0.035/−0.008	<0.001
Quetiapine	−0.053	−0.078/−0.018	<0.001
Risperidone	−0.013	−0.027/0.008	0.210
Valproate	0.005	−0.023/0.043	0.611

**Table 3 ijms-21-08333-t003:** The data for the 15 MANGO neurogenesis genes regulated in the same direction by 3 or more of the drugs.

**Gene**	**Amisulpride**			**Aripiprazole**			**Clozapine**			**Lamotrigine**		
	**logFC**	***p*** **value**	**FDR q**	**logFC**	***p*** **value**	**FDR q**	**logFC**	***p*** **value**	**FDR q**	**logFC**	***p*** **value**	**FDR q**
APOE	−0.40	**3.16 × 10^−5^**	**0.015**	−0.46	**7.31 × 10^−5^**	**0.012**	−0.27	**0.0016**	**0.042**	−0.08	0.13	0.79
CCDC88A	0.11	**0.039**	0.27	0.31	**0.00023**	**0.023**	0.06	0.29	0.62	0.03	0.50	0.96
CNTFR	−0.09	0.29	0.62	−0.21	**0.035**	0.21	0.00	0.96	1.00	0.02	0.70	1.00
CST3	−0.30	**7.07 × 10^−5^**	**0.020**	−0.44	**1.37 × 10^−5^**	**0.0045**	−0.25	**0.00076**	**0.026**	0.00	0.95	1.00
EGR1	−0.08	0.56	0.82	−0.01	0.98	1.00	0.13	0.24	0.57	−0.14	0.31	0.92
EPHB4	−0.22	**0.0012**	0.06	−0.31	**0.00059**	**0.037**	−0.23	**0.0018**	**0.045**	−0.07	0.28	0.90
FABP3	0.06	0.31	0.65	0.04	0.57	0.79	0.51	**7.98 × 10^−5^**	**0.0055**	−0.04	0.49	0.96
GPX1	−0.22	**0.00063**	**0.046**	−0.34	**0.0013**	0.051	−0.22	**0.00066**	**0.023**	−0.07	0.12	0.79
MMP2	−0.13	**0.031**	0.25	−0.28	**0.00034**	**0.028**	−0.18	**0.0013**	**0.037**	−0.07	**0.040**	0.66
NOTCH1	−0.24	**0.0077**	0.14	−0.42	**0.00049**	**0.033**	−0.40	**1.26 × 10^−6^**	**0.00030**	−0.03	0.63	0.99
NOTCH2	−0.06	0.32	0.65	−0.12	0.07	0.28	−0.28	**6.05 × 10^−7^**	**0.00019**	−0.08	**0.016**	0.55
NOTCH3	−0.31	**0.00033**	**0.033**	−0.37	**0.00031**	**0.027**	−0.47	**5.83 × 10^−8^**	**3.95 × 10^−5^**	−0.07	0.11	0.77
NPAS3	−0.09	0.27	0.60	−0.23	**0.030**	0.20	−0.10	0.27	0.60	0.13	0.08	0.73
SOX2	−0.22	**0.0027**	0.09	−0.25	**0.0072**	0.11	−0.24	**0.00021**	**0.010**	0.02	0.69	1.00
SOX3	−0.30	**0.043**	0.28	−0.53	**0.0019**	0.06	−0.69	**6.54 × 10^−6^**	**0.00092**	−0.26	0.09	0.75
**Gene**	**Lithium**			**Quetiapine**			**Valproate**			**Risperidone**		
	**logFC**	***p*** **value**	**FDR q**	**logFC**	***p*** **value**	**FDR q**	**logFC**	***p value***	**FDR q**	**logFC**	***p*** **value**	**FDR q**
APOE	0.03	0.65	0.93	−0.08	**0.048**	0.12	−0.09	0.17	0.36	−0.10	0.18	0.81
CCDC88A	0.21	**0.00061**	**0.022**	0.19	**0.00013**	**0.00087**	−0.06	0.33	0.55	0.12	**0.019**	0.48
CNTFR	0.39	**7.04 × 10^−7^**	**9.84 × 10^−5^**	0.75	**5.58 × 10^−58^**	**5.45 × 10^−55^**	0.38	**2.08 × 10^−6^**	**4.29 × 10^−5^**	0.01	0.95	1.00
CST3	−0.02	0.76	0.96	−0.22	**8.85 × 10^−7^**	**9.94 × 10^−6^**	−0.16	**0.023**	0.09	−0.11	0.07	0.64
EGR1	0.68	**1.31 × 10^−7^**	**2.46 × 10^−5^**	0.93	**1.39 × 10^−17^**	**7.35 × 10^−16^**	0.78	**1.05 × 10^−9^**	**5.42 × 10^−8^**	−0.12	0.31	0.88
EPHB4	0.04	0.48	0.86	−0.10	0.09	0.20	−0.28	**1.18 × 10^−5^**	**0.00019**	−0.07	0.27	0.87
FABP3	0.19	**0.0015**	**0.043**	0.57	**1.54 × 10^−23^**	**1.52 × 10^−21^**	0.26	**1.32 × 10^−5^**	**0.00021**	0.06	0.24	0.85
GPX1	−0.10	0.06	0.39	−0.22	**2.33 × 10^−9^**	**4.31 × 10^−8^**	−0.08	0.17	0.36	−0.16	**0.0040**	0.30
MMP2	−0.04	0.38	0.80	−0.36	**1.01 × 10^−24^**	**1.09 × 10^−22^**	−0.20	**0.00010**	**0.0012**	−0.09	0.07	0.65
NOTCH1	−0.06	0.51	0.87	−0.12	**0.015**	**0.048**	−0.42	**1.69 × 10^−7^**	**4.83 × 10^−6^**	−0.02	0.77	1.00
NOTCH2	−0.02	0.67	0.94	−0.33	**1.41 × 10^−26^**	**1.79 × 10^−24^**	−0.14	**0.0064**	**0.033**	−0.06	0.14	0.77
NOTCH3	−0.03	0.65	0.93	0.04	0.32	0.49	0.12	0.09	0.23	−0.08	0.22	0.83
NPAS3	−0.33	**0.00098**	**0.032**	−0.23	**0.0034**	**0.014**	−0.27	**0.0091**	**0.043**	−0.11	0.23	0.84
SOX2	−0.08	0.30	0.73	−0.31	**1.75 × 10^−8^**	**2.76 × 10^−7^**	−0.22	**0.0036**	**0.022**	−0.09	0.10	0.71
SOX3	−0.49	**0.0018**	**0.048**	−1.00	**9.55 × 10^−9^**	**1.59 × 10^−7^**	−0.74	**2.22 × 10^−6^**	**4.54 × 10^−5^**	−0.11	0.37	0.90

**Table 4 ijms-21-08333-t004:** The effects of quetiapine, clozapine, and valproate on the regulation of 3 target genes of *SOX2*.

	Quetiapine				Clozapine				Valproate			
	logFC	logCPM	*p* Value	FDR	logFC	logCPM	*p* Value	FDR	logFC	logCPM	*p* Value	FDR
DKK1	−0.41	4.21	**2.42 × 10^−6^**	**2.45 × 10^−5^**	−0.10	5.10	0.35	0.68	−0.09	4.83	0.35	0.57
BMP4	−0.41	4.21	**2.15 × 10^−6^**	**2.21 × 10^−5^**	−0.13	4.61	0.19	0.52	−0.24	4.72	**0.009**	**0.044**
CCND1	−0.74	8.69	**4.28 × 10^−52^**	**2.84 × 10^−49^**	−0.37	9.12	**6.84 × 10^−6^**	**9.42 × 10^−4^**	−0.05	9.26	0.36	0.58

**Table 5 ijms-21-08333-t005:** The effects of the drugs on metabolites involved in neurogenesis.

Metabolite	Amisulpride		Aripiprazole		Clozapine		Lamotrigine		Quetiapine		Lithium	
	logFC	*p* Value	logFC	*p* Value	logFC	*p* Value	logFC	*p* Value	logFC	*p* Value	logFC	*p* Value
***N*** **-Acetyl-l-aspartic acid**	**0.55**	**<0.001**	**0.55**	**<0.001**	**0.39**	**<0.001**	**0.08**	**0.48**	**−0.04**	**0.7**	**0.37**	**0.002**
**l** **-Glutamic acid**	**0.35**	**<0.001**	**0.27**	**<0.001**	**0.07**	**0.16**	**0.09**	**0.14**	**−0.05**	**0.29**	**0.26**	**0.003**
**Gamma-Aminobutyric acid**	**0.25**	**0.043**	**0.25**	**<0.001**	**0.07**	**0.015**	**−0.06**	**0.59**	**0.42**	**0.016**	**−0.23**	**0.024**
**Glutathione**	**0.19**	**0.7**	**0.93**	**0.003**	**0.2**	**0.58**	**−0.42**	**0.43**	**−2.94**	**0.002**	**0.78**	**0.024**
**l** **-Glutamine**	**0.29**	**0.026**	**0.34**	**<0.001**	**0.14**	**0.2**	**0.07**	**0.51**	**−0.13**	**0.13**	**0.45**	**<0.001**
**l** **-Lactic acid**	**−0.35**	**0.042**	**0.07**	**0.69**	**0.21**	**0.17**	**0.18**	**0.43**	**0.73**	**<0.001**	**0.11**	**0.27**

**Table 6 ijms-21-08333-t006:** Primers used in this study.

Gene	Forward Primer	Reverse Primer	Efficiency
*BMP4*	GGAGGAGGAGGAAGAGCAGA	TTCTTCGTGGTGGAAGCTCC	99.66%
*CCND1*	GATGCCAACCTCCTCAACGA	GGAAGCGGTCCAGGTAGTTC	96.39%
*CNTFR*	AAGGGCTTCTACTGCAGCTG	CATGTAGCGAATGTGGCAGC	99.57%
*DKK1*	TGGAACTCCCCTGTGATTGC	ATAGGCAGTGCAGCACCTTT	99.60%
*EGR1*	CACCTGACCGCAGAGTCTTT	CTGACCAAGCTGAAGAGGGG	99.24%
*FABP3*	AGAAATGGGACGGGCAAGAG	AATGTGGTGCTGAGTCGAGG	98.10%
*SOX2*	ATGGGTTCGGTGGTCAAGTC	ACATGTGAAGTCTGCTGGGG	99.49%
*SOX3*	GTACAGACCAGGACCGTGTG	TCGGTCAGCAGTTTCCAGTC	99.31%

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
