# Peer review of "Transcriptional Effects of Psychoactive Drugs on Genes Involved in Neurogenesis"

_ijms, 2020, doi:10.3390/ijms21218333_

Round 1

Reviewer 1 Report

The paper by Bortolasci et al. analyzed the effects of psychoactive drugs used in the treatment of affective disorders (bipolar disorder and schizophrenia), on the expression of genes involved in neurogenesis. To this end, the authors used 8 common psychoactive drugs for the treatment of the pluripotent NTera2/cloneD1 (NT2) cell line, derived from human testicular embryonic carcinoma cells and performed Genome wide mRNA expression analysis. Further, they used several bioinformatic tools such as Gene ontology (GO), Mammalian Adult Neurogenesis Gene Ontology (MANGO), and TRRUST database to elucidate the function of the genes, gene products properties, and transcriptional networks. The analyses showed that 6 of the 8 drugs (Clozapine, amisulpride, aripiprazole, lithium, quetiapine, and lamotrigine) significantly decreased the expression of genes involved in neurogenesis. As possible mechanisms by which these drugs decrease neurogenesis, they identified the downregulation of transcription factor SOX2 and its target genes BMP4, CCDN1 and DKK1.

This topic is interesting considering the increasing importance of understanding the response of neurogenic cell populations to stress, psychiatric disorders, and psychoactive drugs in treating perception, mood, consciousness, and behavior. The approaches to screen for the alteration of gene expression and to search for factors causing its change by several drugs are reasonable and well conducted. The interpretation of data analysis is summarized well and the supported by the presented results. However, several questions arose from the data presented. Below are the major issues that should be considered in further improving this manuscript

Major concerns:

  1. The title is not supported by the content of the article. The authors’ data is limited to the observation of genes involved in neurogenesis at the transcriptional level in vitro. The study does not present any data demonstrating indeed an alteration in neurogenesis. Please modify the title accordingly so it better reflects the data presented.

  1. A similar concern is extended to the Discussion section. The authors discuss that the downregulation of BMP4CCDN1 and DKK1 by quetiapine could represent a molecular mechanism for the reduction in neurogenesis observed in this study after treatment with quetiapine. Again, the impact on neurogenesis was not tested in this manuscript. The authors should soften their conclusions when stating an impact on neurogenesis. Additional experiments are required if the authors want to maintain their conclusions on the reduction of neurogenesis by the psychoactive drugs tested. For example, BrdU assays in combination with endogenous neuronal markers should be done. Otherwise it is uncertain whether psychoactive drugs in the study indeed decreased the transition of proliferative and multipotent NSCs to fully differentiated neurons and glia.

  1. The authors focused on the GO neurogenesis as the main part of the manuscript. However, this approach is somewhat biased. GO enrichment analyses list genes associated with biological processes, cellular compartments, and molecular functions. To fully assess the significance of these drugs on neurogenesis the authors need to show the results of other GO term categories and discussed how neurogenesis is represented among the other categories.

  1. The analysis by TRRUST database resulted in SOX2 as a major regulator of the list of MANGO neurogenesis genes. TRRUST will generate a list of transcription factors that could affect the expression of the genes submitted for query. For transparency, the authors need to include a list of all transcription factors generated by the analysis. This could demonstrate SOX2 is a key transcription factor among others affected by psychoactive drugs used.

  1. Additional validation of transcriptional changes is lacking. It is necessary to validate the changes observed in the different gene targets discussed in the manuscript, for example by conducting RT-qPCR or immunoblotting (i.e. protein levels of SOX2 and expression levels of the three major targets. Validation experiments are part of the standard when presenting global transcriptomic analyses and to support any conclusion out of those analyses.

  1. For drug treatments, NT2 cells were differentiated into post-mitotic neurons (NT2-N) following RA treatment. Previously, it was reported that the RA-induced neuronal differentiation of NT2 cells is accompanied by down-regulation of SOX2. The authors need to explain how reduction in SOX2 is exclusively induced by psychoactive drugs and not by RA.

  1. Previous reports also suggested that generation of neurons from NSCs depends on the inhibition of Sox1-3 expression by proneural proteins. The authors suggest that inhibition of SOX2 and its target genes lead to decreasing neurogenesis. The authors need to reconcile these contradicting reports.

  1. It is unclear throughout the manuscript which drugs are indeed those 6 drugs that decreased the expression of genes involved in neurogenesis. Please state clearly in the manuscript the list of those 6 drugs. In addition, Risperidone is described as the only drug that induced neurogenesis, but Figure 1 shows no differentially expressed genes (lane 94-95) and then Table 2 shows a no significant p-value in MANGO. Please clarify what data is used to conclude that Risperidone increases transcription of genes involved in neurogenesis.

  1. The authors need to soften their conclusions on the impact of these drugs on the transcription of genes in neurogenesis and avoid generalization. Their experimental conditions only allow for conclusion when the drugs are used in vitro in an acute dose and for a short period of time (24h). These conditions are often far from the actual use of these drugs in humans. Indeed, the authors comment on the introduction (Lane 66) that chronic treatment of certain atypical antipsychotics induce neurogenesis. No discussion is presented by the effects of long-term treatment in the discussion section and how different dose/treatments can change the impact on neurogenesis.

Minor concerns:

  1. Please consider including the list of some genes presented in each category in Figure 2. Please include a more descriptive legend on Figure 2 to describe the meaning of the width and length of each block.

  1. Studies of antipsychotic treatment in humans are referenced in Discussion section. Please specify which antipsychotics were used. This might enable to explore the physiological effects of drugs used in this manuscript.

  1. Please provide the rationale of the different concentrations used for each drug in the study.

Reviewer 2 Report

As I mentioned in my previous review this research should be  extended for additional studies evaluating neurogenesis. According to the author's informations it is not feasible to add the suggested extra data.

I accept the revised manusccript.

This manuscript is a resubmission of an earlier submission. The following is a list of the peer review reports and author responses from that submission.

Round 1

Reviewer 1 Report

The paper by Bortolasci et al. analyzed the effects of psychoactive drugs used in the treatment of affective disorders (bipolar disorder and schizophrenia), on the expression of genes involved in neurogenesis. To this end, the authors used 8 common psychoactive drugs for the treatment of the pluripotent NTera2/cloneD1 (NT2) cell line, derived from human testicular embryonic carcinoma cells and performed Genome wide mRNA expression analysis. Further, they used several bioinformatic tools such as Gene ontology (GO), Mammalian Adult Neurogenesis Gene Ontology (MANGO), and TRRUST database to elucidate the function of the genes, gene products properties, and transcriptional networks. The analyses showed that 6 of the 8 drugs (Clozapine, amisulpride, aripiprazole, lithium, quetiapine, and lamotrigine) significantly decreased the expression of genes involved in neurogenesis. As possible mechanisms by which these drugs decrease neurogenesis, they identified the downregulation of transcription factor SOX2 and its target genes BMP4, CCDN1 and DKK1.

This topic is interesting considering the increasing importance of understanding the response of neurogenic cell populations to stress, psychiatric disorders, and psychoactive drugs in treating perception, mood, consciousness, and behavior. The approaches to screen for the alteration of gene expression and to search for factors causing its change by several drugs are reasonable and well conducted. The interpretation of data analysis is summarized well and the supported by the presented results. However, several questions arose from the data presented. Below are the major issues that should be considered in further improving this manuscript

Major concerns:

  1. The title is not supported by the content of the article. The authors’ data is limited to the observation of genes involved in neurogenesis at the transcriptional level in vitro. The study does not present any data demonstrating indeed an alteration in neurogenesis. Please modify the title accordingly so it better reflects the data presented.

  1. A similar concern is extended to the Discussion section. The authors discuss that the downregulation of BMP4, CCDN1 and DKK1 by quetiapine could represent a molecular mechanism for the reduction in neurogenesis observed in this study after treatment with quetiapine. Again, the impact on neurogenesis was not tested in this manuscript. The authors should soften their conclusions when stating an impact on neurogenesis. Additional experiments are required if the authors want to maintain their conclusions on the reduction of neurogenesis by the psychoactive drugs tested. For example, BrdU assays in combination with endogenous neuronal markers should be done. Otherwise it is uncertain whether psychoactive drugs in the study indeed decreased the transition of proliferative and multipotent NSCs to fully differentiated neurons and glia.

  1. The authors focused on the GO neurogenesis as the main part of the manuscript. However, this approach is somewhat biased. GO enrichment analyses list genes associated with biological processes, cellular compartments, and molecular functions. To fully assess the significance of these drugs on neurogenesis the authors need to show the results of other GO term categories and discussed how neurogenesis is represented among the other categories.

  1. The analysis by TRRUST database resulted in SOX2 as a major regulator of the list of MANGO neurogenesis genes. TRRUST will generate a list of transcription factors that could affect the expression of the genes submitted for query. For transparency, the authors need to include a list of all transcription factors generated by the analysis. This could demonstrate SOX2 is a key transcription factor among others affected by psychoactive drugs used.

  1. Additional validation of transcriptional changes is lacking. It is necessary to validate the changes observed in the different gene targets discussed in the manuscript, for example by conducting RT-qPCR or immunoblotting (i.e. protein levels of SOX2 and expression levels of the three major targets. Validation experiments are part of the standard when presenting global transcriptomic analyses and to support any conclusion out of those analyses.

  1. For drug treatments, NT2 cells were differentiated into post-mitotic neurons (NT2-N) following RA treatment. Previously, it was reported that the RA-induced neuronal differentiation of NT2 cells is accompanied by down-regulation of SOX2. The authors need to explain how reduction in SOX2 is exclusively induced by psychoactive drugs and not by RA.

  1. Previous reports also suggested that generation of neurons from NSCs depends on the inhibition of Sox1-3 expression by proneural proteins. The authors suggest that inhibition of SOX2 and its target genes lead to decreasing neurogenesis. The authors need to reconcile these contradicting reports.

  1. It is unclear throughout the manuscript which drugs are indeed those 6 drugs that decreased the expression of genes involved in neurogenesis. Please state clearly in the manuscript the list of those 6 drugs. In addition, Risperidone is described as the only drug that induced neurogenesis, but Figure 1 shows no differentially expressed genes (lane 94-95) and then Table 2 shows a no significant p-value in MANGO. Please clarify what data is used to conclude that Risperidone increases transcription of genes involved in neurogenesis.

  1. The authors need to soften their conclusions on the impact of these drugs on the transcription of genes in neurogenesis and avoid generalization. Their experimental conditions only allow for conclusion when the drugs are used in vitro in an acute dose and for a short period of time (24h). These conditions are often far from the actual use of these drugs in humans. Indeed, the authors comment on the introduction (Lane 66) that chronic treatment of certain atypical antipsychotics induce neurogenesis. No discussion is presented by the effects of long-term treatment in the discussion section and how different dose/treatments can change the impact on neurogenesis.

Minor concerns:

  1. Please consider including the list of some genes presented in each category in Figure 2. Please include a more descriptive legend on Figure 2 to describe the meaning of the width and length of each block.

  1. Studies of antipsychotic treatment in humans are referenced in Discussion section. Please specify which antipsychotics were used. This might enable to explore the physiological effects of drugs used in this manuscript.

  1. Please provide the rationale of the different concentrations used for each drug in the study.

Reviewer 2 Report

The aim of the study was evaluation of the impact of some selected antypsychotic drugs on the proces of neurogenesis at a transcriptional level in vitro. mRNA expression was quantified and analysed using Gene set enrichment analysis, with the neurogenesis gene set retrieved from the Gene Ontology database and the Mammalian Adult Neurogenesis Gene Ontology (MANGO) database. Obtained results indicated that 6 of the 8 tested drugs decreased the expression of genes involved in neurogenesis in both databases which, according to the authors, may suggest a negative regulation of the expression of genes involved in neurogenesis. In my opinion it's just the beginning of the study and more advanced at least in vitro (but also in vivo) studies are needed. First of all protein expression should be confirmed with Western blott.  Next step should be evaluation of some selected   neurometabolites involved in the neurogenesis process like Alanine, Aspartate, Creatinine, Glucose, GABA, Glutamate, Glutamine, Glutathione, Lactate, N-Acetyl Aspartate ect after treatment with selected drugs. The results obtained from the analysis the two databases are insufficient to extend unambiguous conclusions regarding the negative impact of selected drugs on neurogenesis.

Author Response

Response to Reviewer 2 Comments

Point 1: The aim of the study was evaluation of the impact of some selected antypsychotic drugs on the proces of neurogenesis at a transcriptional level in vitro. mRNA expression was quantified and analysed using Gene set enrichment analysis, with the neurogenesis gene set retrieved from the Gene Ontology database and the Mammalian Adult Neurogenesis Gene Ontology (MANGO) database. Obtained results indicated that 6 of the 8 tested drugs decreased the expression of genes involved in neurogenesis in both databases which, according to the authors, may suggest a negative regulation of the expression of genes involved in neurogenesis. In my opinion it's just the beginning of the study and more advanced at least in vitro (but also in vivo) studies are needed. First of all protein expression should be confirmed with Western blott.  Next step should be evaluation of some selected   neurometabolites involved in the neurogenesis process like Alanine, Aspartate, Creatinine, Glucose, GABA, Glutamate, Glutamine, Glutathione, Lactate, N-Acetyl Aspartate ect after treatment with selected drugs. The results obtained from the analysis the two databases are insufficient to extend unambiguous conclusions regarding the negative impact of selected drugs on neurogenesis.

Response 1: Thank you for your feedback, we agree that further experiments would enrich the findings. However, it is not feasible to add the suggested extra data since we only have 10 days to revise the manuscript. We do agree that additional experiments would be useful to validate the transcriptional effects that we observed but cannot complete such experiments in the time frame provided.

Round 2

Reviewer 2 Report

I received a response form the authors with the explanation that  they can't add the suggested extra data since they only have 10 days to revise the manuscript. I have already noted in the first review that the results obtained from the analysis just two databases are insufficient to extend unambiguous conclusions regarding the negative impact of selected drugs on neurogenesis. In a situation, where the authors do not follow the suggestions, I can't accept this work because it will be contrary to my previous decision.  I support my opinion that the presentation of results based on two databases is not enough for publication in the International Journal of Molelular Science and addtional experiments should be performed.